# Brief communication: The crucial assessment of possible significant vertical movements preceding the 28 December 1908, $M_W$=7.1, Messina Straits earthquake

Nicola Alessandro Pino[1]

[1]Istituto Nazionale di Geofisica e Vulcanologia, Sezione di Napoli, via Diocleziano 328, Napoli, 80124, Italia

*Correspondence to*: N. A. Pino (alessandro.pino@ingv.it)

**Abstract.** The 28 December 1908 Messina Straits earthquake represents one of the worst seismic catastrophes in whole world history. In spite of the relatively large and various amount of data available and of the geophysical investigations accomplished in the Straits, the seismogenic structure is still elusive and intensely debated. Some models proposed for the causative fault rely considerably on the hypothesis of significant subsidence preceding the event. By driving results that differ critically from previously debated models, this assumption would have crucial repercussions on the seismic hazard assessment in the area. However, the critical analysis of this assumption in the light of the available data allows the rejection of this possibility.

## 1 Introduction

On 28 December 1908, at 5:20, a tremendous shake awoke the population in the Messina Straits area, in Southern Italy (Fig. 1). Messina and Reggio Calabria, the two cities located on the facing sides of the Straits, with population respectively of ~140,000 and ~45,000, were almost completely destroyed by the strong ground shaking and the following tsunami. The death toll was assessed between 60,000 and more than 100,000, with a most probable estimate of 80,000, making this earthquake one of the most deadly seismic events in the whole human history and certainly the most catastrophic ever in Europe. In spite of the numerous scientific analyses published since then providing a considerable variety model faults (see Pino et al., 2009 and references therein), after more than 115 years no convincing seismogenic fault has been detected by geophysical analyses (Argnani, 2021) and the location, geometry, and kinematics of the causative fault are still matter of discussion among scholars. The Straits area, considered one of highest seismic hazard in Italy according to the current estimate (http://esse1-gis.mi.ingv.it/), is also frequently in the spotlight because of the decades-long debate on the project of a ~3 km-long bridge connecting Messina and Reggio Calabria, which presently is back on the table of the Italian national government, which is currently evaluating single- and multi-span designs along with the possibility of (https://www.mit.gov.it/comunicazione/news/ponte-messina-mims-avviata-la-procedura-per-la-realizzazione-di-uno-studio-di). These elements keep encouraging further investigations searching for evidences of possible seismogenic faults capable of M=7.1 earthquakes in the Messina Straits. Indeed, in addition to the numerous studies in the literature, in the last few years several other articles have been published proposing distinct model faults, characterized by very different location and

geometry. In the most recent one (Barreca et al., 2021), in addition to the sudden sinking of large portion of the Straits area caused by the 28 December 1908 earthquake, the authors hypothesize a significant subsidence (0.4 m) along the coasts of Sicily and Calabria, preceding the seismic event, as a consequence of aseismic slip on a low angle, east-dipping fault, and their resulting coseismic model fault is strongly dependent on this assumption. Actually, some precursory subsidence occurring in the two years before the earthquake was previously deduced also by other authors (Mulargia and Boschi, 1983; Baldi et al.,

1983; Bottari et al., 1992), from tide gauge data.

Although the 1908 earthquake occurred at the dawn of the quantitative seismology, several data type are available to investigate its source characteristics. In fact, reliable information could be determined on both the seismic moment and the rupture directivity from historical seismograms (Pino et al., 2000), and the felt intensity observations have been used to discriminate among the various proposed model faults (e.g., Convertito and Pino, 2014). However, all the most robust, quantitative models

for the 1908 causative fault have been deduced from the analysis of the available historical levelling data.

As obvious, by using these data the seismic moment and the resulting location, geometry, and kinematics of the coseismic dislocation would be strongly affected by the hypothesis of significant preseismic vertical ground motion overlapped to the earthquake effects in the data. This, in turn, would also have critical consequences on the evaluation of the seismic hazard in the Messina Straits area, where now live more than 700,000 people (the cities of Messina and Reggio Calabria total more than

400,000), making the possible subsidence preceding the 1908 earthquake a crucial issue, for either the determination of the causative fault or the assessment of possible precursory elements, deserving careful consideration. In this note, I face this subject, by critically analyzing the relevant historical data.

## 2 Critical discussion of the available data

The available suitable data to investigate the possible occurrence of vertical movement in the Messina Straits area preceding

the 1908 earthquake are i) the levelling measurements made across the time of the event and ii) the tide gauge data gathered in the Messina harbor during a few decades including the quake.

### 2.1 Levellng data

In 1909, right after the 28 December 1908 earthquake, the Istituto Geografico Militare (Italian Military Geographic Institute) remeasured two levelling lines along the coasts of the sides of the Straits (Fig. 1; Loperfido, 1909). The survey revealed that

the shore subsided almost everywhere along the two facing shores relative to the previous survey, with maximum value of 0.58 m on the Calabrian side and 0.71 m on the Sicilian side. As reference point, Loperfido (1909) clearly states that the measures on the Sicilian shore are referred to a point (Colle S. Rizzo, located about 5 km inland) "for which there was no doubt", i.e., it was far enough, "in a region not perturbed by the last earthquake, thus remained unchanged". Nevertheless, some others (Barreca et al., 2021) neglect all the Sicilian benchmarks because – they say – the referring point "is probably too

close to the area affect by the subsidence". Some other authors (e.g., Boschi et al., 1989) prefer not to include the measures in

the Messina harbor area in their analysis, based on the notes by Loperfido (1909) who notice that some of these benchmarks might have suffered collapse.

The line on the Sicilian side had been previously surveyed in 1898-1899, while the Calabrian side was remeasured starting in 1907 and ending in December 1908. Thus, if any subsidence had to be recorded by the levelling data it should have happened during these time spans. In principle, If the whole area underwent subsidence throughout the decade preceding the earthquake, with different rate along the two shores, the line in Calabria would have recorded only a part of it and there is no chance to determine univocally from the levelling data how much of it occurred on the Sicilian side during 1898-1907 and how much in Calabria during 1907-1908. Similarly, if the same subsidence rate is assumed for the two sides of the Straits, it will not be possible to separate the two processes neither to solve reliably for the sum of pre- and coseismic motion from the levelling data. By considering only one of the two lines, the source resulting from the modeling is to be considered an overlap of the precursory vertical movement – occurring in the time span between the survey and the previous reference measurement – and the coseismic effects. Therefore, by considering only the Calabrian line, as done for instance by Barreca et al. (2021), if any significant subsidence preceding the 1908 earthquake overlaps the coseismic effects, it must have occurred in a time significantly shorter than two years (subsidence rate larger than 0.2 m/y) and should have had significant effects, visible by eye.

Not secondarily, the 1908 earthquake seismic moment $M_0$ determined geodetically by considering only coseismic vertical ground movements is well consistent with the value derived from the analysis of the historical seismograms (Pino et al., 2000), while the consideration of substantial preseismic subsidence would necessarily lower the geodetic $M_0$.

## 2.2 Tide gauge data

The "Mati-Ricci" tide gauge installed at the Messina harbor was damaged by the 1908 earthquake and tsunami, and was restored in April 1909. Except for this period, the height of the sea level at this location has been regularly measured from January 1897 to February 1923. Together with the measures from the other Italian tide gauges, the data from the Messina harbor were reported as monthly and yearly relative height of the mean sea level, in the "Processo verbale delle sedute della R. Commissione Geodetica Italiana", published once a year by the Italian Geodetic Commission. Successively, they were also included in the Monthly and Annual Meah Heights of Sea-Level (Association D'Oceanographie Physique, 1940) and are currently available from the Permanent Service for Mean Sea Level (PSMSL, http://www.psmsl.org/data/obtaining/met.monthly.data/115.metdata). Figure 2 displays a plot of the original values reported in the "Processo verbale delle sedute della R. Commissione Geodetica Italiana".

No failure of these data is reported, hence their reliability cannot be questioned. In these regards, it is worth mentioning that the Messina tide gauge web page at PSMSL reports the warning "This is not research quality data. Use with extreme caution". However, it should be noted that the scope of the PSMLS is the long-term sea level change information, as explicitly specified on their webpage; this sentence is their standard warning for all sites where there are doubts about the long-term stability of

the records and, in the case of the historical Messina data, it is there to warn users about the presence of the significant effects of the 1908 earthquake in the time series, as referred by PSMLS office on specific inquiry (personal email communication).

As for the meaning of the reported values, in general, the sea level was measured with respect to a benchmark. At the Messina harbor site, the measures represent the distance of the sea level, with unit of 1 m and positive axis directed downward, from a reference point was a metal benchmark placed in the gauge-house, thus above the sea level (Association D'Oceanographie Physique, 1940), like for most Italian tide gauges at that time. As a consequence, larger (smaller) positive values correspond to lower (higher) sea levels.

The fundamental information on the orientation of the axis has been often overlooked, with resulting major ambiguities. In fact, Loperfido (1909) first report the Messina harbor sea level data for years 1897-1908, in form of annual average, without any other specification on which direction had to be taken as positive. Thus, although Omori (1913) explicitly states that for Messina tide gauge data are referred to the benchmark, meaning that "the larger figures correspond to the lower levels and the smaller figures to the higher levels", Mulargia and Boschi (1983) assume that larger (smaller) positive values represent higher

(lower) sea level (i.e., opposite to the actual orientation) and deduce "uplift in the coast at a rate of 16.2 mm/y in the period before 1900, a downlift of 2.8 mm/y in the period 1900-1906 and a rapid "sinking" of the coast at 22.7 mm/y in the 24 months before the earthquake". Baldi et al. (1983) reach the same conclusion and, a few years later, Bottari et al. (1992) make the same mistake and observe that "the epicentral area of the Messina earthquake experienced subsidence (from 1902)".

By taking into account the correct sign of the sea level variations, in the two years preceding the 1908 earthquake no ground

subsidence at all results at the Messina harbor relative to the sea level and, considering the above discussion on the reliability of the tide gauge data, this is an unambiguous and well constrained conclusion. Indeed, for this period the measures indicate a slight relative ground uplift, erroneously considered as subsidence by some authors not assuming the right orientation of the axis.

Incidentally, it is worth noticing that the proper orientation of the sea level variation allows accounting for the slow postseismic

subsidence detected for several years after the earthquake, which corresponds to postseismic viscous relaxation of the lower crust (Cannelli et al., 2013), as often observed for considerable crustal seismic events (e.g., Pino, 2012).

## 3. Concluding remarks

The two cities located on the facing shores of the Straits, Messina and Reggio Calabria, have always been in very tight connection with the sea, being based on marine activities (fishing, shipping, recreational beach attendance, …), even with train

ferries going back and forth across the Straits. As examples, Figure 3 displays some views of the Messina and Reggio Calabria shore before the 1908 earthquake, showing the proximity of the daily life to the sea in the two cities, with either commercial or recreational activities occurring along the shoreline. With such small elevations above the water, if a permanent vertical movements of tens of centimetres ever occurred, particularly if it developed in a few months, people's actions would have suffered macroscopic consequences, causing some concern at any rate and, thus, very much likely leaving some sign in the

records. Not even a single official report or newspaper short article mentioning such an event has ever been found (Comerci et al., 2015; Guidoboni et al., 2019). Albeit not quantitative, this observation further makes the assumption of precursory subsidence unjustified.

As a conclusion, it can be definitely stated that no hypothesis of significant vertical movement preceding the 1908 earthquake can be considered reliable and, in turn, fault models relying on this assumption cannot be considered acceptable.

Overall, based on the available data some constraints can be put on the causative fault of the 28 December 1908 earthquake. By jointly considering the published results of geodetic, seismic instrumental, and macroseismic analyses (see Pino et al., 2009, for a comprehensive review), the most likely source corresponds to an about 40 km-long fault, roughly N-S oriented and dipping eastward at low angle. The rupture should have nucleated at the southern end of the Straits, at 8-12 km depth, and propagated northward, as confirmed by the seismograms' analysis (Pino et al., 2000) and the modelling of the macroseismic
data (Convertito and Pino, 2014). These characteristics appear to be quite robust; nevertheless, in principle, future investigations could demonstrate their fallacy, but whatever criticism should be grounded on solid elements and rigorous analyses, rather than unfounded hypotheses.

Apparently, the above elements represent the best constrained indications that can be derived from measurements dating more than one century ago. On the other hand, the many geophysical prospections carried out in the Messina Straits did not
succeeded in finding the fault. Maybe it is time to start thinking of specific investigations, such as drilling the upper crust looking for the fault.

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

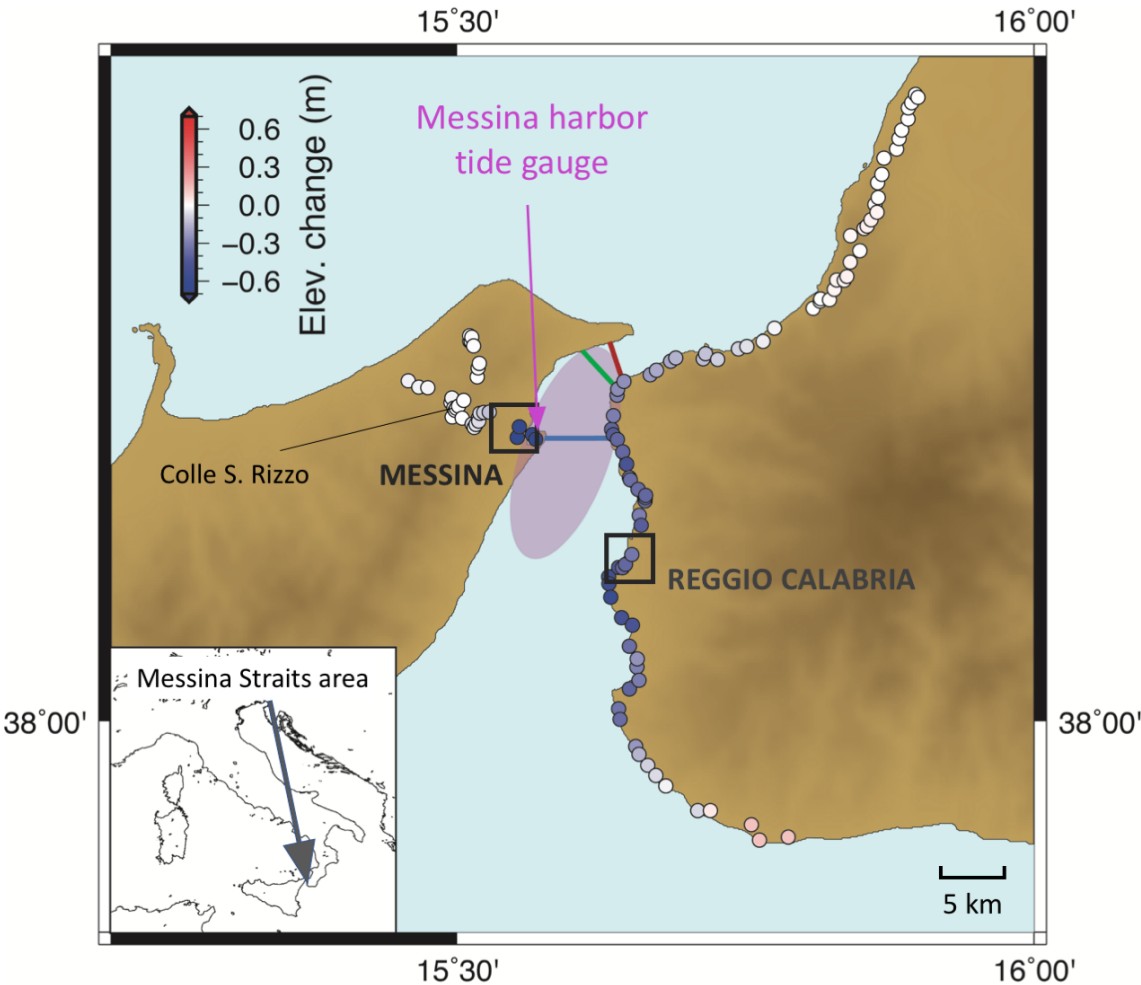


**Figure 1: Messina Straits area. The filled circles correspond to the benchmarks of the levelling lines (Loperfido, 1909), with the color scale indicating the measured elevation change while the pink shaded oval represents approximately the area of the 1908 earthquake precursory subsidence larger than 0.4 m, according to Barreca et al. (2021). The segments across the shores of the Straits indicate**

**the track of the alternative projects of connection between Sicily and Calabria, presently under evaluation: single-span bridge (red), multi-span bridge (green), and undersea tunnel (blue). The figure was partly generated with Generic Mapping Tool (GMT; Wessel and Smith, 1991).**

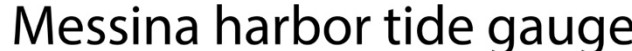

## Messina harbor tide gauge

28 December 1908
earthquake

monthly average ——
annual average ——

Relative sea level (m)

Time (yr)

200

**Figure 2: Monthly (black) and annual (only for the 1897-1908; red) mean height of sea level recorded at the Messina harbor tide gauge. The positive direction of the vertical axis is downward, as was the case for the tide gauge, with the measures representing the height of the sea level relative to the benchmark located in gauge-house, above the sea level.**

205

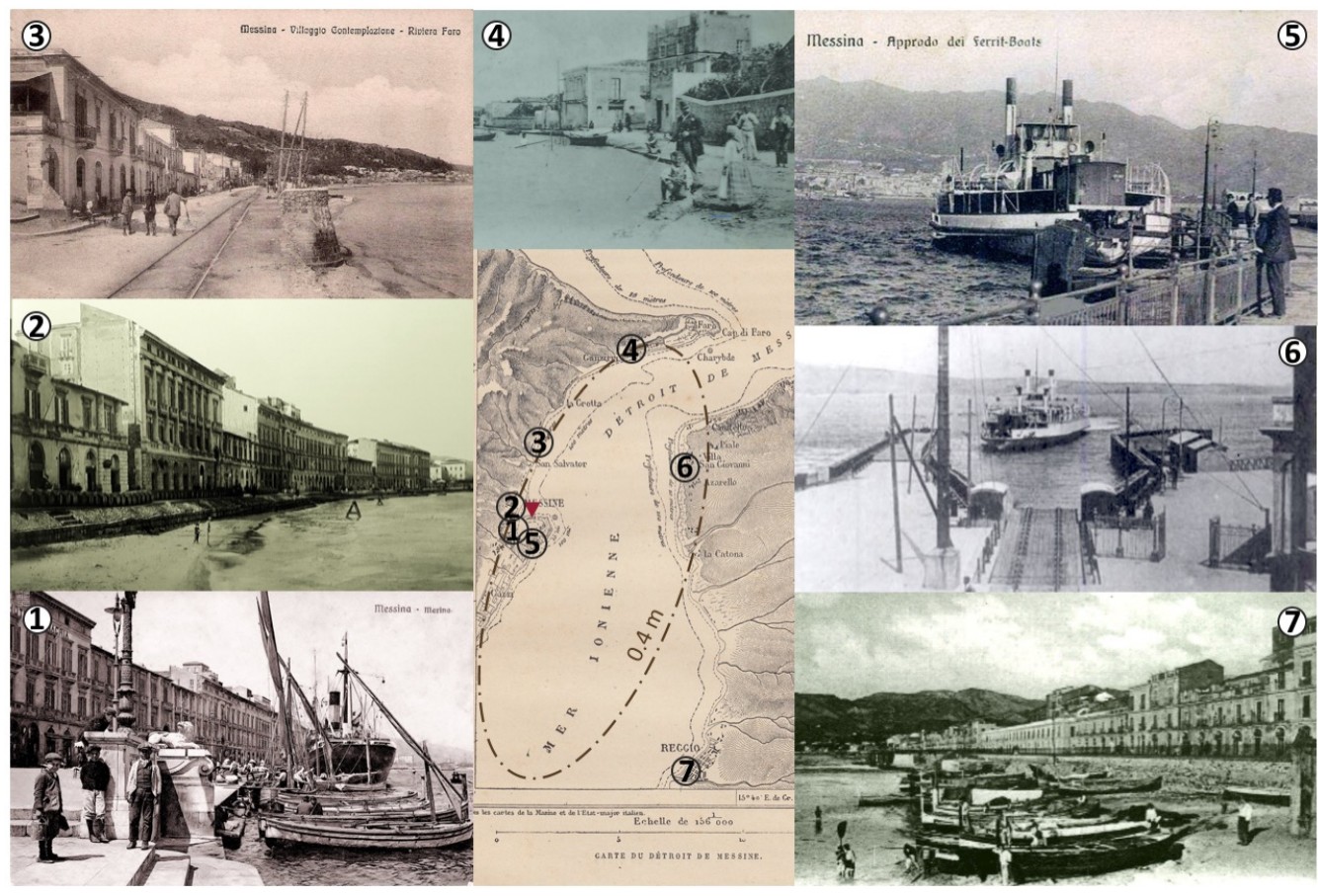

**Figure 3: Images of the shoreline at several locations (relevant numbers on the map) in Messina and Reggio Calabria. All the photographs date back to few years preceding the 1908 earthquake. The dashed ellipse marks approximately the 0.4 m subsidence isoline according to Barreca et al. (2021). If such a macroscopic vertical change ever occurred, it would have had significant consequences on people daily life, along the Messina Straits shores.**