# Peer review of "Brief communication: The crucial assessment of possible significant vertical movements preceding the 28 December 1908, M=7.1, Messina Straits earthquake"

_Natural Hazards and Earth System Sciences, 2022_

## Referee Comment (RC1)

The manuscript represents an original contribution to the scientific discussion on the M= 7.1, Messina, 1908 earthquake, occurred in the Messina Strait area. It was the worst catastrophic seismic event occurred in Italy and the scientific discussion is still very lively, as far as the assessment of the seismic source and the definition of the causative fault. This point could contribute to the knowledge of the seismic hazard of that area, therefore has a real significance as far as possible consequences on people living in the area, on industrial activity and on infrastructures. Moreover, the project of a ~3 km-long bridge across the Strait, connecting Reggio Calabria and Messina, is back on the table of the Italian national government, with all the scientific studies concerning the definition of seismic vulnerability of the project.

The manuscript carries out a logical analysis on the available data on the vertical movements inferred for both cost lines before the earthquake, by analyzing levelling and tidal gauges published data. The Author cites most of the previous papers published on this subject and particularly on the vertical costal movements inferred from levelling and tidal gauges data. He critically analyzed different conclusions and data available and made his own considerations giving a possible interpretations of the vertical movements before the earthquake.

The paper is concise and very well written and clear to understand.

I am here asking if it is possible to have the Author's interpretation concerning the features of the causative fault of the earthquake based on the results of this short communication, if he can contribute to the scientific debate by adding some consequences of his analysis in terms of assessment of the causative fault of the Messina earthquake.

In other terms, if he could add something to this sentence "As a conclusion, it can be definitely stated that no hypothesis of significant vertical movement preceding the 1908 earthquake can be considered reliable and, in turn, fault models relying on this assumption cannot be considered acceptable."

---

## Referee Comment (RC3)

[referee-annotated manuscript omitted]

---

## Author Response (AR1)

**Point-by-point reply to the reviewers' comments**

**Reviewer #1**

**The manuscript represents an original contribution to the scientific discussion on the M= 7.1, Messina, 1908 earthquake, occurred in the Messina Strait area. It was the worst catastrophic seismic event occurred in Italy and the scientific discussion is still very lively, as far as the assessment of the seismic source and the definition of the causative fault. This point could contribute to the knowledge of the seismic hazard of that area, therefore has a real significance as far as possible consequences on people living in the area, on industrial activity and on infrastructures. Moreover, the project of a ~3 km-long bridge across the Strait, connecting Reggio Calabria and Messina, is back on the table of the Italian national government, with all the scientific studies concerning the definition of seismic vulnerability of the project.**

**The manuscript carries out a logical analysis on the available data on the vertical movements inferred for both cost lines before the earthquake, by analyzing levelling and tidal gauges published data. The Author cites most of the previous papers published on this subject and particularly on the vertical costal movements inferred from levelling and tidal gauges data. He critically analyzed different conclusions and data available and made his own considerations giving a possible interpretations of the vertical movements before the earthquake.**

**The paper is concise and very well written and clear to understand.**

**I am here asking if it is possible to have the Author's interpretation concerning the features of the causative fault of the earthquake based on the results of this short communication, if he can contribute to the scientific debate by adding some consequences of his analysis in terms of assessment of the causative fault of the Messina earthquake.**

**In other terms, if he could add something to this sentence "As a conclusion, it can be definitely stated that no hypothesis of significant vertical movement preceding the 1908 earthquake can be considered reliable and, in turn, fault models relying on this assumption cannot be considered acceptable."**

*As also stated below in response to the Reviewer #2 comments, my note is not meant to be a review of the proposed fault models for the 1908 earthquake nor to provide new tectonic interpretations. However, I definitely agree that a few words about the constraints put by the well-assessed coseismic geophysical observation on the source of the 1908 earthquake might help in framing the data discussed in the note in the general debate. For this reason, I had in the Concluding remarks the following sentences:*

"Overall, based on the available data some constraints can be put on the causative fault of the 28 December 1908 earthquake. By jointly considering the published results of geodetic, seismic instrumental, and macroseismic analyses (see Pino et al., 2009, for a comprehensive review), the most likely source corresponds to an about 40 km-long fault, roughly N-S oriented and dipping eastward at low angle. The rupture should have nucleated at the southern end of the Straits, at 8-12 km depth, and propagated northward, as confirmed by the seismograms' analysis (Pino et al., 2000)

and the modelling of the macroseismic data (Convertito and Pino, 2014). These characteristics appear to be quite robust; nevertheless, in principle, future investigations could demonstrate their fallacy, but whatever criticism should be grounded on solid elements and rigorous analyses, rather than unfounded hypotheses.

Apparently, the above elements represent the best constrained indications that can be derived from measurements dating more than one century ago. On the other hand, the many geophysical prospections carried out in the Messina Straits did not succeeded in finding the fault. Maybe it is time to start thinking of specific investigations, such as drilling the upper crust looking for the fault."

**Reviewer #2**

**In my opinion it is very clear what Pino highlights in this brief communication. Subsidence processes claimed by Barreca et al. 2021 should have been recorded in somehow. Levelling data cannot be used to separate the two signals of co-seismic and pre- earthquake subsidence. Furthermore, tide gauge graph shows that there is no relative subsidence recorded before the earthquake, suggesting that the claimed modelled subsidence should be ruled out, as the result of some sort of aseismic slipalong this E- dipping low angle discontinuity, beneath the supposed W-Fault. Tide gauge shows a slightly uplifting area as confirmed by preserved marine terraces, resulting from a long- term uplift process (see the recent review in Meschis et al., 2019; 2022; for instance). About the subsidence measured by levelling data in the Sicilian side, it would be simpler to think that benchmarks are located invery steep slopes affected by landslides processes and highly fractured rock formations, suggesting that those ones are not reliable to record any co-seismic movement. This latter is also shown by Comerci et al. (2015; 2020) as well as no evidence of pre-earthquake subsidence is reported.**

**It is important to note that evidence that the W-Fault tectonically deforms the seafloor is actually a confirmation that active faulting is occurring offshore. Indeed, the trace of W-Fault in its offshore part more or less coincides with the one proposed by Doglioni et al. (2012; Doglioni, C., Ligi, M., Scrocca, D., Bigi, S., Bortoluzzi, G., Carminati, E., ... & Riguzzi, F. (2012). The tectonic puzzle of the Messina area (Southern Italy): Insights from new seismic reflection data. Scientific Reports (1), 1-9.) and later used by Meschis et al. (2019). In the fault model used by Meschis et al. (2019) there is no pre-earthquake subsidence claimed, instead a co-seismic uplift is calculated matching the area affected by long-term "footwall uplift" shown by tectonically-deformed marine terraces (Meschis et al., 2022; Meschis, M., Roberts, G. P., Robertson, J., Mildon, Z. K., Sahy, D., Goswami, R., ... & Iezzi, F. , 2022, Out of phase Quaternary uplift-rate changes reveal normal fault interaction, implied by deformed marine palaeoshorelines. Volume 416, 1 November 2022, 108432), with lower rates of uplift closer to the fault tips in Messina and Taormina towns and higher ones in centre of the fault, closer to Roccalumera village.**

**Finally, a prominent pre-earthquake subsidence claimed as high as 0.2 m/yr should have been noted by people living and working by the sea at that time. This is ruled aout by the detailed analysis of historical sources for the 1908 earthquake available in the literaure**

**(e.g., Comerci et al., 2015; 2020; and reference therein).**

**SPECIFIC COMMENT**

**Reference to the papers cited above should be included in the manuscript. Minor specific comments are included in the annotated manuscript attached here**

*Most of the Reviewer's comments are relative to interpretations or discussion of structural and/or tectonic issues. I would like to make clear that in my brief note I deal with geophysical observations regarding a relatively short time preceding the 1908 earthquake and I prefer not to enter any discussion about interpretations of structural and/or tectonic issues (e.g., Doglioni et al., 2012) or discuss proposed possible fault models (e.g., Meschis et al., 2019). My main points are: i) no official report mention any subsidence, as correctly stressed by the Reviewer; ii) the Messina harbor tide gauge data, often misinterpreted in the past and leading to the conclusion that some subsidence might have occurred, clearly show that no precursory subsidence preceded the 1908 earthquake, instead. Any comment on possible long-time tectonic uplift relative to faults for which no structural evidence has been provided and whose existence is uncertain (e.g., Argnani, 2012, ESR) is far beyond the scope of the note.*

*In this regard, I consider the suggestion about the reference Comerci et al. (2015) very appropriate and useful and I will definitely add this article to the list, while Comerci et al. (2020) is essentially a version in Italian of the previous paper. As for the coseismic macroseismic effect, in the present version of the manuscript I also added the reference of the CFTI (Catalogo dei forti terremoti in Italia; Guidoboni et al., 2019).*

*As for the other suggested references, I underline that my note is not meant to be a review paper, and I am not listing the environmental effects, with the relevant references. I just mentioned the analysis by Convertito and Pino (2014) – a quantitative modeling of felt report data – to stress that most analysis of the 1908 earthquakes have been based on the historical geodetic data, but felt report data can be reliably used to deduce quantitative information about the source.*

**Minor comments annotated on the manuscript**

***please make reference to more recent papers for a review on this topic***

*The reference reported in the introduction, Pino et al. (2009) – which to date still is the most complete and detailed review of the very many (tens) articles published in the first century after the earthquake – is only mentioned to say that in spite of the large number of analyses accomplished on this issues, the fault still remains elusive and new studies are published every year. Successive publications do not include such a comprehensive list.*